# DAVE: A VLM Vision Encoder for Document Understanding and Web Agents

**Brandon Huang[1,2], Hang Hua[1], Zhuoran Yu[1,3], Trevor Darrell[2],**
**Rogerio Feris[1,†], and Roei Herzig[1,†]**

[1]MIT-IBM Watson AI Lab, [2]UC Berkeley, [3]University of Wisconsin–Madison

## Abstract

While Vision–language models (VLMs) have demonstrated remarkable performance across multi-modal tasks, their choice of vision encoders presents a fundamental weakness: their low-level features lack the robust structural and spatial information essential for document understanding and web agents. To bridge this gap, we introduce DAVE, a vision encoder purpose-built for VLMs and tailored for these tasks. Our training pipeline is designed to leverage abundant unlabeled data to bypass the need for costly large-scale annotations for document and web images. We begin with a self-supervised pretraining stage on unlabeled images, followed by a supervised autoregressive pretraining stage, where the model learns tasks like parsing and localization from limited, high-quality data. Within the supervised stage, we adopt two strategies to improve our encoder's alignment with both general visual knowledge and diverse document and web agentic tasks: (i) We introduce a novel model-merging scheme, combining encoders trained with different text decoders to ensure broad compatibility with different web agentic architectures. (ii) We use ensemble training to fuse features from pretrained generalist encoders (e.g., SigLIP2) with our own document and web-specific representations. Extensive experiments on classic document tasks, VQAs, web localization, and agent-based benchmarks validate the effectiveness of our approach, establishing DAVE as a strong vision encoder for document and web applications. Code available at `https://github.com/Brandon3964/DAVE`.

## 1 Introduction

Vision-Language Models (VLMs) (Liu et al., 2024a; Bai et al., 2025; Achiam et al., 2023) have shown remarkable capabilities in multi-modal reasoning and understanding, enabling a wide range of applications from image captioning to interactive web agents (Wu et al., 2024; Qin et al., 2025; Liu et al., 2023; Li et al., 2022b). A central component of VLMs is the vision encoder (Yin et al., 2024), which converts images into visual tokens for the language backbone to process (Li et al., 2023b; Liu et al., 2023). As such, the design of these encoders has become a key focus in VLM research (Tong et al., 2024; Shi et al., 2024).

However, the vision encoders predominantly used by VLMs suffer from a fundamental weakness: their low-level features lack the robust structural and spatial information essential for document understanding and web agents (Tschannen et al., 2025; Radford et al., 2021). Conversely, DINO-style models (Oquab et al., 2023) offer low-level features yet are tuned to natural images (Oquab et al., 2023; Siméoni et al., 2025) and transfer poorly to documents, UIs, and diagrams (Tong et al., 2024). To bridge this gap, we introduce DAVE (***Document and web Agents Vision Encoder***), a vision encoder purpose-built for VLMs and tailored for these tasks. See Figure 1 for an overview.

A key challenge in training on document and web images is the scarcity of high-quality annotated data, since current annotations rely on OCR models that bottleneck both scalability and quality (Kim et al., 2021; Lee et al., 2023). We address this with a two-stage training process: self-supervised pretraining on large-scale unlabeled data, followed by autoregressive pretraining on limited annotated structural and grounding data. Yet, the supervised stage introduces a further challenge: specialized

---

†Equal advising. Correspondence to: `zhaobin@berkeley.edu`, `hang.hua1@ibm.com`

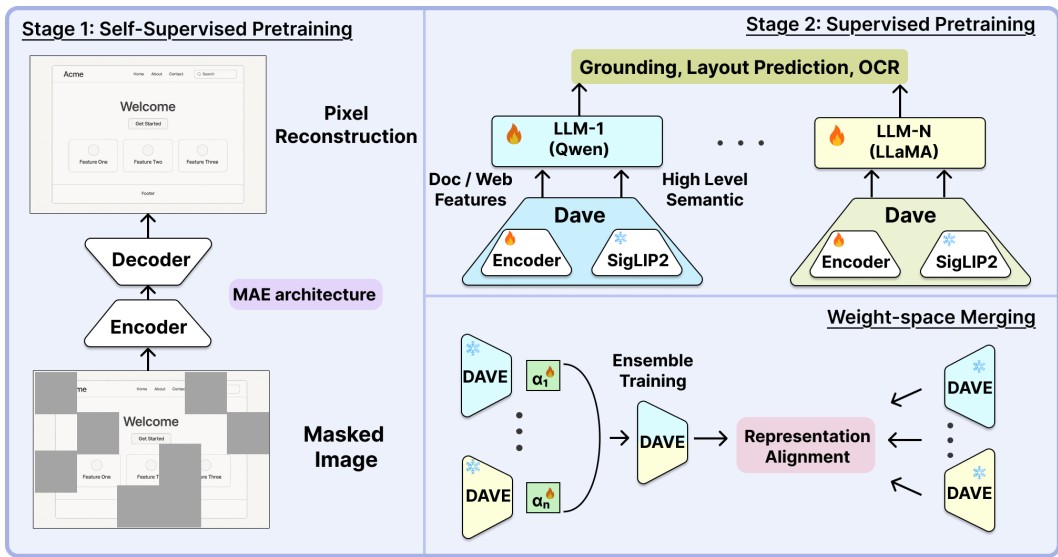

Figure 1: **DAVE Overview**. Stage 1 trains the vision encoder with a decoder using MAE, learning strong structural and spatial priors from unlabeled data. Stage 2 performs autoregressive pretraining on diverse tasks with different text decoders and fuses the high-level semantic features from SigLIP 2. After that, different encoders are combined into a single one by learning a merge coefficient using unsupervised representation alignment, while keeping the encoders frozen.

vision encoders trained with a single text decoder tend to overfit to that decoder, leading to misalignment when paired with other decoders. We mitigate this issue via multi-decoder pretraining and weight-merging ("model soup") (Wortsman et al., 2022), which produces a largely decoder-agnostic encoder. To preserve high-level semantics while leveraging our structural and spatial features, we fuse a generalist encoder with our encoder during pretraining via ensembling. The resulting specialized encoder integrates seamlessly with diverse decoders and generalist VLM stacks. Finally, our purpose-built vision encoder serves as the core component of a VLM architecture specifically designed for document and web agent tasks. We evaluate DAVE across diverse settings, including traditional and vision–language tasks. On traditional benchmarks, DAVE surpasses all SOTA models in document recognition and segmentation, while achieving competitive results with SigLIP 2 on screenshot classification. For vision–language evaluation, we test on document understanding, web localization, and web agent. Notably, **DAVE improves performance by an average of 10.5% compared to SigLIP 2**, highlighting its ability to provide specialized representations. Finally, we evaluate web agent settings on the Mind2Web benchmark (Deng et al., 2023), where DAVE improves web agent performance by 5% over the strongest baseline vision encoder.

We summarize our main contributions as follows: **(i)** We introduce DAVE, a vision encoder purpose-built for VLMs and tailored for document and web agents tasks; **(ii)** We propose a two-stage pretraining framework that combines a self-supervised pretraining stage on unlabeled images, followed by a supervised autoregressive pretraining stage; **(iii)** We introduce model weight merging and ensemble training strategies to enhance the alignment of structural and spatial representation for documents and web images with diverse VLM and agentic frameworks; **(iv)** We perform extensive experiments to evaluate the impact of visual representations on document and web understanding, as well as downstream web-agent tasks.

## 2 RELATED WORK

**Pretrained Vision Encoders**. Image–text contrastive learning (Radford et al., 2021; Zhai et al., 2023; Jia et al., 2021; Yu et al., 2022) and self-supervised representation learning (He et al., 2022; Zhou et al., 2021; Oquab et al., 2023; Chen et al., 2020; Caron et al., 2021) are the two main paradigms for large-scale visual pretraining. Recent work has sought to combine the strengths of the two approaches through joint training or post-hoc alignment (Tschannen et al., 2025; Fini et al.,

2025; Maninis et al., 2024; Naeem et al., 2024), yielding strong encoders for both vision and multimodal tasks. The works most closely related to ours are Eagle (Shi et al., 2024) and Perception Encoder (Bolya et al., 2025), which align pretrained vision encoders to a pretrained LLM with large-scale general datasets. In contrast, we pretrain the vision encoder from scratch with self-supervised learning, use multiple pretrained LLMs as a backbone, and use domain-specific data to create a specialized foundation model.

**Document Understanding with LLMs**. Document processing and understanding have transitioned from OCR-based (Xu et al., 2020b;a) methods to vision-language-based. One line of work trains encoder-decoder models with various objectives, like extracting structures (Feng et al., 2025; Kim et al., 2022; Lv et al., 2023; Lee et al., 2023), and image reconstruction conditioned on text (Huang et al., 2022; Tang et al., 2023). Recent advancements in VLMs led to another line of work that uses document data to finetune a VLM (Nassar et al., 2025; Hu et al., 2024; Liu et al., 2024g) or incorporate it as part of the VLM training (Team et al., 2025; Bai et al., 2025; Chen et al., 2024). Our work seeks to bridge these two lines of work by introducing a specialized vision encoder within a general VLM framework.

**Vision-Language Models**. Inspired by the success of recent large language models (LLMs) (Brown et al., 2020; Dubey et al., 2024), Vision-language models (Radford et al., 2021; Liu et al., 2024a; Sun et al., 2025; Hua et al., 2024b; Ye et al., 2023; Hu et al., 2022; Tang et al., 2024; Bai et al., 2025; Zhu et al., 2025; Hua et al., 2025a; Tong et al., 2024; Avogaro et al., 2026) aim to achieve multimodal intelligence by jointly understanding and generating visual and language information. Flamingo (Alayrac et al., 2022) and BLIP-2 (Li et al., 2023b) are two of the early works that explore integrating LLMs as part of VLM. Beginning with LLaVA (Liu et al., 2024a), researchers have used instruction-following chat data in VQA format for instruction tuning, achieving significantly improved results (Liu et al., 2024e; Li et al., 2023a; Hua et al., 2024c;a; Tang et al., 2025; Hua et al., 2025b; Lu et al., 2023).

**Multimodal Web Agents**. Vision-based autonomous web agents have recently attracted attention for their simplicity and stronger generalization compared to LLM-based agents (Deng et al., 2023; Kim et al., 2023). Early efforts such as WebGUM (Furuta et al., 2023) and CogAgent (Hong et al., 2024) pretrained vision–language models (VLMs) on web and GUI data to enhance agentic capabilities. Follow-up work (Wu et al., 2024; Xu et al., 2024; Liu et al., 2024b; Qin et al., 2025) expanded training with large-scale grounding and interaction datasets, further improving VLM performance on web-based tasks. While these works are promising steps toward a general-purpose visual web agent, the role of the vision encoder in these systems remains underexplored.

## 3 THE DAVE MODEL

### 3.1 PRELIMINARILIES

**Masked Autoencoder**. Masked Autoencoder (MAE) (He et al., 2022) adopts an asymmetric encoder–decoder architecture. The vision encoder processes only a subset of visible patches, while a lightweight transformer decoder reconstructs the full image from the encoded features and mask tokens. After pretraining, only the encoder is retained for downstream tasks.

By default, MAE applies per-patch normalization before computing the reconstruction loss to learn better representations. The reconstruction loss is then defined as

$$\mathcal{L}_{\text{MAE}} = \frac{1}{|\mathcal{M}|} \sum_{i \in \mathcal{M}} \left\| f_\theta(\tilde{x})_i - \frac{x_i - \mu(x_i)}{\sqrt{\sigma^2(x_i) + \epsilon}} \right\|_2^2, \quad \epsilon = 10^{-6} \tag{1}$$

where $\tilde{x}$ denotes the input image with masked patches, $f_\theta(\tilde{x})_i$ is the reconstructed output of patch $i$ predicted by the decoder with parameters $\theta$, $x_i$ is the ground-truth pixel values of patch $i$, $\mu(x_i)$ and $\sigma^2(x_i)$ are the mean and variance of pixels within patch $i$, $\mathcal{M}$ is the set of masked patches.

**Vision–Language Model**. A vision–language model (VLM) consists of a vision encoder $\phi$, an MLP projector, and a text decoder $\Theta$. The vision encoder takes an input image $x \in \mathbb{R}^{H \times W \times 3}$ and produces a sequence of patch-level features $\{v_i\}_{i=1}^N$. These are projected into the text embedding space via the MLP projector and used along with the text input.

In what follows, we describe the process to get our vision encoder.

## 3.2 Stage 1: SSL on Document and Web Images

In Stage 1, we conduct self-supervised training with MAE to learn rich structural and spatial features from document and web images. While MAE demonstrates the strongest performance on OCR tasks compared to other methods, the training is prone to instability when trained at scale (Fan et al., 2025). We find that this instability is particularly acute for document and web images. Our analysis (Section 5.4) attributes this to their characteristically low inter-patch variance, which destabilizes the target normalization in the standard MAE objective (Equation 1). To address this, we modify the objective to reconstruct raw pixel values directly:

$$\mathcal{L}_{\text{MAE-pixel}} = \frac{1}{|\mathcal{M}|} \sum_{i \in \mathcal{M}} \| f_\theta(\tilde{x})_i - x_i \|_2^2, \tag{2}$$

This change stabilizes training and enables scaling the training sample to 20 million images without additional hyperparameter tuning. With this robust self-supervised procedure, we then proceed to the second stage of our training process.

## 3.3 Stage 2: Supervised Multi-Task Pretraining

To further enhance the encoder's structural and spatial understanding for document and web images, we perform a supervised, autoregressive pretraining stage. This stage utilizes a limited set of high-quality labeled data for tasks like OCR, layout extraction, and web localizations. We use the VLM architecture as described in Section 3.1, and DAVE as the vision in the VLM architecture.

**Weight-space Merging**. A key limitation of the architecture discussed above is that the pretrained vision encoder becomes tightly coupled to its specific text decoder. This coupling significantly degrades performance when the encoder is integrated with a different decoder. To address this, we use model merging to create a vision encoder that is agnostic to the choice of text decoder.

Formally, given a set of $n$ pretrained text decoders $\{\Theta_1, \ldots, \Theta_n\}$, we train $n$ corresponding instances of our vision encoder, denoted $\{\phi_1, \ldots, \phi_n\}$. Each instance is aligned with a different text decoder but otherwise shares the same architecture, training data, and hyperparameters.

To merge these different encoders, we propose a distillation-based merging scheme that learns a small set of coefficients to combine pretrained weights while keeping the original parameters frozen. Considering each encoder $\phi_i$ as a set of $m$ weights $\{\theta_i^{(j)}\}_{j=1}^m$, we form each merged weight $\theta_{\text{merge}}^{(j)}$ by learning a set of coefficients $\{\alpha_i^{(j)}\}_{i=1}^n$ to compute the weighted sum of the corresponding weight for each of the $n$ encoders:

$$\theta_{\text{merge}}^{(j)} = \sum_{i=1}^n \alpha_i^{(j)} \theta_i^{(j)} \qquad \alpha_i^{(j)} \in [0, 1].$$

The merged encoder is thus composed of $m$ merged weights: $\phi_{\text{merge}} = \{\theta_{\text{merge}}^{(j)}\}_{j=1}^m$.

The resulting encoder produces patch-level features $\mathbf{z} = \phi_{\text{merge}}(\mathcal{I})$. To ensure that $\phi_{\text{merge}}$ preserves the features from all $n$ teacher encoders, we define a distillation loss, $\mathcal{L}_{\text{distill}}$. This objective minimizes the average Mean Squared Error between the merged features $\mathbf{z}$ and the features from each teacher encoder $\mathbf{z}_i = \phi_i(\mathcal{I})$:

$$\mathcal{L}_{\text{distill}} = \frac{1}{n} \sum_{i=1}^n \| \hat{\mathbf{z}}_i - \mathbf{z}_i \|_2^2.$$

During the distillation process, all encoder parameters remain frozen; only the newly introduced combination coefficients are optimized. The final vision encoder is given by the merged encoder:

$$\phi_{\text{DAVE}}^{\text{final}} = \phi_{\text{merge}}(\{\alpha_i^\star\}_{i=1}^n),$$

where $\{\alpha_i^\star\}_{i=1}^n$ are the optimized coefficients obtained from this training.

The next section details our method for fusing specialized document and web features with general visual representations.

**Ensemble Training**. Pretraining the encoder exclusively on document and web data provides strong structural and spatial features, but it also limits its grasp of general visual representation. This is a

crucial shortcoming, as the high-level semantic features learned from diverse, large-scale datasets are equally important for robust performance.

To obtain both types of features, we design an ensemble pretraining paradigm that combines a frozen pretrained generalist encoder $\phi_{\text{gen}}$ with our document and web specialist encoder $\phi_{\text{spec}}$ from the previous stage: $\phi_{\text{DAVE}}(x) = \text{Concat}\big(\phi_{\text{gen}}(x),\ \phi_{\text{spec}}(x)\big)$.

This design provides two main benefits: (i) it encourages $\phi_{\text{spec}}(x)$ to focus on learning low-level structural and spatial representations, as high-level semantics are captured by $\phi_{\text{gen}}(x)$; and (ii) it enables early fusion of structural and spatial features with high-level semantic features.

## 4 EVALUATION

### 4.1 IMPLEMENTATION DETAILS

**Self-supervised Stage**. For the SSL stage, we follow the original MAE implementation (He et al., 2022), with ViT-L16-384 (Dosovitskiy et al., 2020) as our vision encoder. We train the MAE with a batch size of 4096 for 120K steps. More detail can be found in Appendix A.1

**Supervised Training Stage**. We adopt the VLM architecture discussed in Section 3.1, with an image tilting size of 384. During the ensemble training, we employ frozen SigLIP-2 as the generalist vision encoder, while the encoder from the Self-supervised stage serves as the domain-specialized component. This forms the full DAVE encoder. We experiment with multiple LLMs of varying scales and architectures, including QWen2.5-0.5B-Instruct (Bai et al., 2025), Phi-4-mini-Instruct (Abouelenin et al., 2025), and Granite-3.1-3B-Instruct (Team et al., 2025). After a single epoch of full-parameter training, we retain only the DAVE encoder for weight-merging and downstream tasks. During the weight-merging distillation, we train the merge coefficient for 20 epochs on unlabeled documents and web images. Refer to Appendix B for implementation details.

### 4.2 EVALUATION SETTING

In this section, we first describe our encoder baselines, followed by two types of evaluations for the vision encoders: (i) finetuning vision encoders for classic document tasks, (ii) performing instruction tuning to build a VLM with the vision encoder for vision-language tasks and agentic tasks.

**Baselines**. We compare our vision encoder with both generalist and specialist encoders. For generalist encoders, we utilize DinoV2 (Oquab et al., 2023) and Web-SSL MAE (Fan et al., 2025), a MAE variant trained on 2 billion images. We also include SigLIP2 (Tschannen et al., 2025) and AIMv2 (Fini et al., 2025), which are SOTA encoders trained with both contrastive and reconstruction. For specialist vision encoders, we compare against DiT (Li et al., 2022a), Pix2Struct (Lee et al., 2023), and Dolphin (Feng et al., 2025). Since both Pix2struct and Dolphin are encoder-decoder models, we use their encoders for comparison.

**Classic Document Tasks**. For each classic document task, we finetune the vision encoder and the suitable prediction heads. In DocBank, we use attention pooling to pool the visual feature, followed by an MLP head to predict the bounding box. In Doclaynet, we train an MLP head to predict the semantic segmentation based on the feature patches. For RICO-SCA, we use attention pooling followed by an MLP to predict the class. Refer to Appendix C for more details.

**Vision-Language Model as Evaluator**. To evaluate our vision encoders on more complicated and realistic tasks, such as VQA and instruction-based localization, we use VLMs as evaluators. Specifically, we follow the standard LLaVA architecture as discussed in Section 3.1. We use a tilting size of 336 for AIMv2 and 384 for all other vision encoders. For vision encoders that produce a different number of visual tokens than the tilting resolution, we use bilinear interpolation as in LLaVA-Onevision (Li et al., 2024) to interpolate the visual tokens. Note that we omit the projector alignment phase to improve training efficiency, as prior work has shown that this stage yields only limited gains Tong et al. (2024); Karamcheti et al. (2024). We use Llama-3.2-3B-Instruct (Dubey et al., 2024) and Qwen2.5-7B-Instruct (Bai et al., 2025) as the LLM backbone. We train for one epoch with the vision encoder frozen. For Mind2web, we finetune the VLMs on the training set before performing offline evaluation on the test set. More detail can be found in Appendix D.

| Model / Variant | Document | | | | | General VQA | | | Web | | |
| --- | --- | --- | --- | --- | --- | --- | --- | --- | --- | --- | --- |
| | AI2D | OCRBench | DocVQA | InfoVQA | ChartQA | MMMU | RealWorldQA | TextVQA | VisualWeb | Screenspot-V2 | WebSRC |
| **Llama-3.2-3B-Instruct** | | | | | | | | | | | |
| MAE-Web | 53.0 | 27.6 | 43.8 | 26.2 | 43.8 | 29.9 | 48.6 | 35.1 | 36.9 | 51.1 | 46.8 |
| DinoV2 | 53.2 | 2.6 | 13.7 | 20.4 | 14.1 | 35.0 | 49.4 | 13.7 | 13.2 | 16.5 | 17.6 |
| DiT | 49.9 | 2.1 | 11.3 | 19.2 | 12.6 | 28.8 | 43.9 | 10.0 | 9.2 | 2.5 | 14.9 |
| Pix2Struct | 51.0 | 6.9 | 22.8 | 21.4 | 21.5 | 33.3 | 45.0 | 16.4 | 22.6 | 32.6 | 26.5 |
| Dophine | 50.8 | 44.7 | 74.8 | 37.4 | 60.3 | 33.3 | 44.4 | 42.8 | 48.8 | 56.3 | 76.0 |
| AIMv2 | 56.1 | 48.3 | 56.6 | 35.6 | 48.7 | 36.3 | 51.4 | 58.7 | 46.7 | 32.0 | 49.4 |
| SigLIP 2 | 58.0 | 51.5 | 72.1 | 40.6 | 51.8 | **36.9** | 53.5 | 64.4 | 54.7 | 40.7 | 67.8 |
| **DAVE (ours)** | **59.6** | **62.2** | **82.1** | **50.2** | **63.1** | 36.6 | **55.6** | **69.2** | **59.2** | **64.5** | **82.6** |
| **Qwen-2.5-7B-Instruct** | | | | | | | | | | | |
| MAE-Web | 63.8 | 33.2 | 55.1 | 28.6 | 50.3 | 38.4 | 45.4 | 40.6 | 59.5 | 70.1 | 60.7 |
| Pix2struct | 58.7 | 10.0 | 29.8 | 20.0 | 28.3 | 37.9 | 39.1 | 18.2 | 52.3 | 47.8 | 30.3 |
| Dophine | 63.3 | 50.9 | 83.3 | 42.0 | 66.3 | 40.2 | 44.6 | 49.4 | 60.7 | 75.2 | 85.6 |
| AIMv2 | 69.7 | 69.3 | 89.3 | 57.8 | 77.4 | 41.3 | 56.2 | 73.1 | 64.9 | 79.3 | 86.1 |
| SigLIP 2 | 73.6 | 67.4 | 90.2 | 55.2 | 75.9 | 43.9 | **58.7** | **74.3** | 65.4 | 81.4 | 88.3 |
| **DAVE (ours)** | **74.0** | **67.5** | **90.9** | **60.2** | **82.5** | **45.8** | 55.2 | 73.7 | **67.3** | **82.9** | **88.6** |

Table 1: The DAVE's performance on Document understanding, general VQA, and Web understanding benchmarks with two VLM architectures using different LLMs. The best result per row is highlighted in **bold** and the second best with underline. Higher values represent better performance.

## 4.3 BENCHMARKS

In this section, we describe our evaluation datasets for several downstream benchmarks.

**Classic Document Understanding Tasks**. To directly evaluate the structural and spatial visual representation from the vision encoders, we consider several document parsing tasks that can be performed with only images. We employ DocBank (Li et al., 2020a) to comprehensively evaluate document recognition on different categories, including tables, charts, and paragraphs. We also use Doclaynet (Pfitzmann et al., 2022) to test document segmentation. For web tasks, we use RICO-SCA (Li et al., 2020b) to perform web UI classification.

**Document and General VQA**. For document understanding and question answering, we evaluate on AI2D (Kembhavi et al., 2016), OCRBench (Liu et al., 2024f), DocVQA (Mathew et al., 2021), InfoVQA (Mathew et al., 2022), and ChartQA (Masry et al., 2022). For general vision-language understanding, we report results on MMMU (Yue et al., 2024), RealWorldQA (xAI, 2024), and TextVQA (Singh et al., 2019).

**Web UI and Agent Tasks**. For web UI localization, we use Screenspot-V2 (Wu et al., 2024), which consists of text instructions (e.g., "click the button in cooridnate (x, y)"), and the corresponding bounding boxes. We report the center accuracy: the label-box center lies within the predicted box. We also evaluate on WebSRC (Chen et al., 2021) and VisualWebBench (Liu et al., 2024d) for web UI question answering. For the agentic benchmark, we evaluate on Mind2Web (Deng et al., 2023), where the model receives a user goal and a webpage screenshot, and must predict an action like clicking.

## 4.4 DATASETS

In this section, we provide a description of our training datasets for the different stages.

**Self-supervised Learning Data**. We use 20 million images for our self-supervised training. For document data, we sample 10 million PDF images from DocFM (Team et al., 2025), which contains 85 million document pages extracted from Common Crawl, Wikipedia, and ESG (Environmental,

| Model | Mind2Web | | | | | |
| | Cross-Task | | Cross-Website | | Cross-Domain | |
| | Element Acc. | Step SR | Element Acc. | Step SR | Element Acc. | Step SR |
|---|---|---|---|---|---|---|
| Llama-3.2-3B-Instruct | | | | | | |
| MAE | 21.0 | 16.8 | 16.5 | 11.9 | 15.5 | 11.6 |
| DinoV2 | 13.5 | 11.4 | 10.2 | 6.6 | 9.5 | 7.2 |
| DiT | 5.8 | 4.1 | 2.7 | 1.2 | 2.1 | 1.1 |
| Pix2Struct | 15.8 | 12.3 | 12.1 | 8.2 | 10.8 | 7.5 |
| Dolphin | 24.0 | 19.6 | 20.9 | 13.6 | 19.3 | 14.5 |
| AIMv2 | 14.7 | 11.6 | 9.2 | 5.9 | 8.8 | 6.1 |
| SigLIP 2 | 20.6 | 17.3 | 11.8 | 8.7 | 12.8 | 9.7 |
| **DAVE (ours)** | **30.8** | **26.1** | **24.2** | **18.0** | **23.9** | **19.1** |

Table 2: **Results on Web Agent.** Performance on Mind2Web with three splits (Cross-Task, Cross-Website, Cross-Domain). We report the stepwise accuracy (correct grounding) and the element accuracy (correct grounding and action) for each task.

Social, and Governance) reports. The PDFs are filtered to include English only. For web screenshot, we sample 10 million from Common-Web (com, 2023) without any language filtering.

**Supervised Learning Data**. For the autoregressive supervised training, we leverage data from diverse sources, including PlotQA (Methani et al., 2020), ChartQA (Masry et al., 2022), fintabnet (Zheng et al., 2021), Datikz (Belouadi et al., 2023), Pubtables (Smock et al., 2022), and DocFM (Team et al., 2025). Importantly, we use the version from Granite Vision (Team et al., 2025) where all the problems are reformulated into content extraction. The data for this stage is a mixture of curated benchmarks and a large-scale, self-processed dataset. The curated benchmarks cover tasks such as chart-to-markdown, table-to-caption, and web UI grounding with UGround Gou et al. (2024). To expand on this, we processed 500K PDF samples from arXiv with an OCR model (Cui et al., 2025), formulating them into additional recognition and grounding tasks. Altogether, this training stage comprises approximately 2 million samples.

**Instruction Tuning Data**. Following Cambrian-1 (Tong et al., 2024), We use the LLaVA-1.5-mix 665K (Liu et al., 2023), DocVQA (Mathew et al., 2021), ChartQA (Masry et al., 2022), and AI2D (Kembhavi et al., 2016) as our instruction tuning data. We also integrate Pixmo-Doc (Yang et al., 2025) into the data mix to improve document understanding. Additionally, we add MultiUI (Liu et al., 2024c) to improve the web and agentic capability. This combined to a total of 2.5 million training images.

| Model | DocLayNet | DocBank | RICO-SCA |
|---|---|---|---|
| DinoV2 | 68.4 | 38.3 | 85.6 |
| MAE-Web | 64.6 | 44.5 | 88.3 |
| Dolphin | 53.8 | 50.5 | 88.4 |
| Pix2Struct | 56.7 | 47.2 | 90.1 |
| AIMv2 | 70.5 | 42.4 | 91.3 |
| SipLIP 2 | 70.8 | 51.7 | **93.3** |
| **DAVE** | **74.1** | **56.9** | 92.8 |

Table 3: Performance comparison on **classic document tasks**. DocLayNet and DocBank use mAP, while RICO-SCA uses classification accuracy.

## 5 RESULTS

### 5.1 VISION-LANGUAGE TASKS

Table 1 shows the evaluation results for document, general, and web benchmarks. Overall, DAVE consistently outperforms the strongest baseline, SigLIP 2 in the Llama-3.2-3B-Instruct setup on 8 document and web benchmarks by an average of 10.5%. This is achieved without losing the general VQA capabilities, like MMMU and RealWorldQA, which implies that the ensemble training successfully merges the structural and spatial features with the general visual features. In addition, when using Qwen-2.5-7B-Instruct as the VLM decoder, we show that the improvements in benchmarks hold a similar trend. This suggests that our merging scheme effectively aligns with different text decoders. Finally, one interesting observation is that specialized models like Pix2Struct and

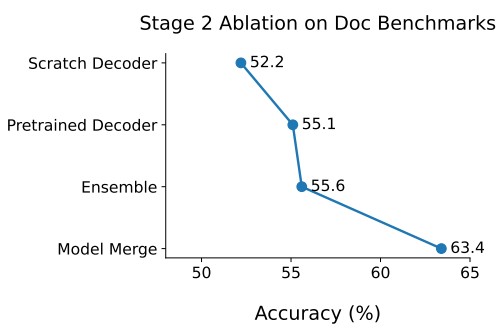

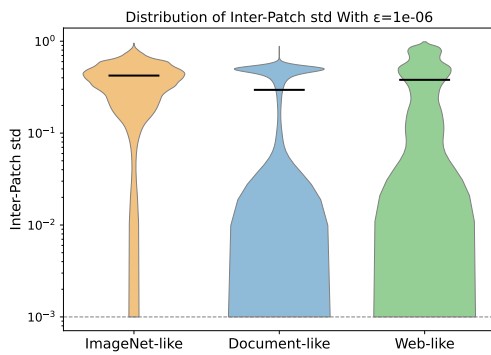

Figure 2: Each row indicates an additional modification to the training strategy.

Figure 3: Inter-patch standard deviation across different data sources.

Dolphin perform suboptimally compared to SigLIP 2 and AIMv2, which shows that both general visual features and document/web-specific features are critical to the performance. This hypothesis is further supported by DAVE's strong results.

## 5.2 WEB AGENTIC TASKS

The results for multimodal-mind2web are outlined in Table 2. DAVE outperforms the best baseline encoder, Dolphin, by an average of 5%. Surprisingly, self-supervised vision encoders like MAE and DinoV2 perform competitively with SigLIP 2 and AIMv2, while document-specialized encoder Dolphine achieves the best accuracy among all baselines. This suggests that structural and spatial aspects may be more important in web navigation compared to broad general features.

## 5.3 CLASSIC DOCUMENT AND WEB TASKS

As shown in Table 3, we evaluate DAVE on dense document tasks. DAVE outperforms both specialized and generalist encoders on DocBank and Doylaynet. This demonstrates the strong structural and spatial visual features learned by our 2-stage pretraining. Furthermore, DAVE performs slightly worse than SigLIP 2 on screen classification, a semantics-heavy task that is not the primary optimization target of DAVE. In principle, DAVE should be able to match SigLIP 2 because it fuses SigLIP 2 features, and additional ablations (See Appendix E.4) confirm that adding SigLIP 2 improves performance, supporting the claim that it contributes high-level semantic information. We attribute DAVE's suboptimal performance compared to SigLIP 2 to the increased hidden dimensionality. Since DAVE's embedding dimension is twice as large as that of SigLIP 2, it is more difficult to pool and process the information with the same one-layer MLP prediction head.

## 5.4 ABLATIONS

In this section, we present detailed ablations on the design choice of our training method. In all of the ablations, we report the average performance on document and web vision-language benchmarks, which we denote as Doc and Web in the Table 4.

**Training Method**. To assess the contribution of each design choice in the stage 2 supervised pretraining, we incrementally add each choice and plot the performance trajectory on document vision-language tasks tasks in Figure 2. Beginning with the a scratch text decoder and subsequently incorporating a pretrained LLM as the text decoder, ensemble training, and finally weight merging. The consistently improving trend highlights the positive impact of our proposed design choices.

**Image Inter-Patch Variance**. To show the instability of the MAE training on document and web image discussed in Section 3.3, we plot the distribution density of inter-patch variance across different data sources in Figure 3. Compared to ImageNet (Deng et al., 2009), document and web images exhibit much lower variance. This distributional gap underscores the need for a specialized

vision encoder capable of handling such data. Additional implementation details are provided in Appendix E.

| Method | Doc | Web |
|---|---|---|
| No merge | 55.6 | 53.0 |
| Average | 62.8 | 67.7 |
| Fisher Merge | 60.3 | 67.0 |
| Learned Coef | 63.4 | 68.2 |

(a) Comparison of Different Merge Methods.

| Method | Doc | Web |
|---|---|---|
| From Scratch | 52.2 | 54.7 |
| Granite | 55.6 | 53.0 |
| Granite+Qwen | 62.1 | 64.8 |
| Granite+Qwen+Phi | 63.4 | 68.2 |

(b) Comparison of Different Merging LLMs.

| Method | Doc | Web |
|---|---|---|
| SIgLIP 2 + DiT | 50.3 | 47.6 |
| SIgLIP 2 + PS | 48.9 | 42.2 |
| SIgLIP 2 + DP | 49.7 | 44.0 |
| DAVE | 63.4 | 68.2 |

(c) Comparison of Different Specialized Encoder Fusion. Key: PS-Pix2struct, DP-Dolphin

| Method | Doc | Web |
|---|---|---|
| SigLIP 2 | 49.1 | 54.4 |
| Finetune SigLIP 2 | 58.2 | 65.2 |
| DAVE | 63.4 | 68.2 |

(d) Comparison with Finetuning.

Table 4: **Ablation study**: (a) merge methods, (b) LLM merging, and additional analyses (c, d).

**Different Merge Setting**. As described in Section 3.3, we use weight merging to create a decoder-agnostic vision encoder. Table 4a examines the effect of different merge strategies. Our end-to-end approach for learning merge coefficients outperforms both simple averaging and the heuristic-based Fisher Merge (Matena & Raffel, 2022) method, which weights parameters by their estimated importance using the Fisher information. Table 4b evaluates merging DAVE from different LLM backbones, where performance improves consistently with the number of merged encoders.

**Multi-encoder Comparison**. Because DAVE combines with both generalist and specialist features, we ask how DAVE performs compared to multi-encoder settings. Specifically, we concatenate different document and web specialized models with SigLIP 2 during the instruction tuning of building the VLM. In Table 4c, we present the result of different multi-encoder settings. We highlight that without any feature fusion, all the specialized encoders fail to improve performance on document and web benchmarks.

**Comparison with Finetuning**. To test whether our pretraining approach is more effective than directly finetuning an existing vision encoder on the supervised data used in our pretraining, we finetune SigLIP 2 with our supervised pretraining data. In Table 4d, we show that DAVE outperforms finetuned SigLIP 2, demonstrating the effectiveness of our approach.

## 6 CONCLUSION

We introduced DAVE, a foundational vision encoder for document and web understanding. Our approach combines self-supervised learning on large-scale unlabeled data with supervision from limited vision–language annotations, enabling data-efficient training in specialized domains. By leveraging pretrained generalist encoders, DAVE can focus on learning domain-specific features. Extensive experiments demonstrate that strong visual representations are critical for document, web, and agentic tasks, where our encoder achieves significant improvements over prior models. Ablation studies further confirm the method's flexibility on different VLM and agentic frameworks.

## 7 LIMITATIONS AND FUTURE WORK

Despite the significant progress achieved in document and web understanding by DAVE, several limitations remain. The model operates at a fixed resolution, depends on an external pretrained generalist encoder, and requires substantial compute for self-supervised training. In agentic settings,

its vision encoder is also not conditioned on prior actions. These limitations highlight opportunities for developing vision encoders that are more compute- and data-efficient, can process inputs of any resolution or aspect ratio, and incorporate prior agentic actions to produce more tailored representations. In addition, this paradigm offers a promising path for domains with scarce vision–language data, such as medical imaging. While we do not anticipate specific negative impacts, as with any machine learning method, caution should be exercised during deployment.

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

| Hyperparameter | Value |
|---|---|
| Batch Size | 4096 |
| Learning Rate | 1e-5 |
| Epochs | 25 |
| Warmup Epochs | 5 |
| Weight Decay | 0.05 |
| Mask Ratio | 0.75 |
| LR Scheduler | Cosine |

Table 5: Training hyperparameters for MAE training.

# A  STAGE 1: SELF-SUPERVISED TRAINING

Here, we present the implementation detail and dataset used for the self-supervised training.

## A.1  IMPLEMENTATION DETAILS

We use the official implementation of masked autoencoder (MAE) for our training, and we use 32 H200 GPUs for the training. Our vision encoder is a ViT-L-384 initialized from scratch, whereas our decoder is a transformer model with a depth of 4 and 16 attention heads. The detail training hyperparameter can be found in Table 5

## A.2  DATASETS

Our self-supervised pretraining dataset consists of 10 million document images from DocFM and 10 million web screenshots from Common Screen

**DocFM**. DocFM is a large scale document data collected by IBM consisting of 85 million document pages extracted from unique PDF documents sourced from Common Crawl, Wikipedia, and ESG (Environmental, Social, and Governance) reports. From this dataset, we first filter and keep only the English pdfs, then we randomly sample 10 million pdfs and store them as images.

**Common Screen**. Common Screen is a large scale web screenshot data consisting of 70 million screenshot images based on the Common Crawl data. We randomly sample 10 million images as our training data, without any language filtering.

## A.3  ADDITIONAL RESULTS ON MAE

| Model | DocBank | Doclaynet | RICO-SCA |
|---|---|---|---|
| MAE-Web | 64.6 | 44.5 | 88.3 |
| MAE-Doc | 72.8 | 52.5 | 90.7 |

Table 6: Evaluation results on classic document tasks.

In this section, we demonstrates the effectiveness of pretraining on document and web images by comparing our stage-1 encoder, which we denote as MAE-Doc, with the the pretrained MAE-Web introduced in 4.2 in classic document tasks. In Table 6, the superior performance of MAE-Doc over MAE-Web highlights importance of the domain-specific pretraining for document and web tasks.

## A.4  TRAINING DIVERGENCE

In Figure 4, we show that training MAE on document and web screenshot images with the normalized pixel loss leads to training divergence.

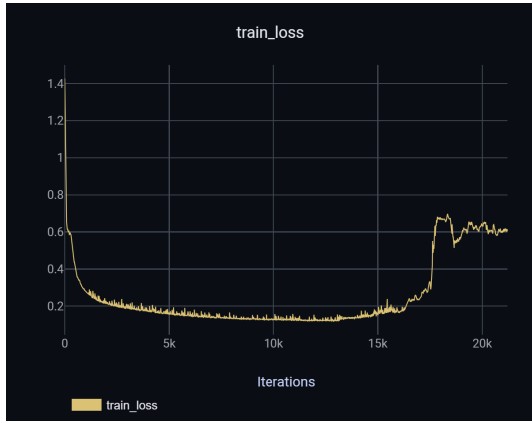

Figure 4: Training loss curve of MAE with normalized pixel as objective.

| Hyperparameter | Value |
|---|---|
| Max sequence length | 20000 |
| Learning Rate | 3e-5 |
| Epochs | 1 |
| Warmup ratio | 0.03 |
| LR Scheduler | Cosine |

Table 7: Training hyperparameters for supervised pretraining.

# B  STAGE 2: SUPERVISED PRETRAINING

In this section, we present the detail implementation and the dataset composition of our supervised pretraining after the self-supervised pretraining.

## B.1  IMPLEMENTATION DETAILS

We use the official LLaVA-Next (Li et al., 2024) repo with some customization to support different LLM and encoder settings, and we use 32 H200 GPUs for the training. To handle high resolution images, we use AnyRes image tilting with size of 384x384. The hyperparameters are listed in Table 7.

## B.2  DATASETS

| Dataset | #Image | Description |
|---|---|---|
| FM4D | 250k | Doc/Chart/Table Extraction |
| PlotQA | 146k | Plot Extraction |
| Fintabnet | 88k | Table Extraction |
| ChartQA | 17k | Chart Extraction |
| Datikz | 94k | Image to Latex |
| Pubtables | 480k | Table to HTML |
| UGround | 756k | Web UI Grounding |
| Self-curated Data | 250k | Doc Grounding/structure Recognition |

Table 8: Dataset description for supervised pretraining.

We use a wide range of publicly available data for our supervised pretraining. Notably, all the data excluding UGround are reformulated into layout and information extraction. For more detail on the data curation, refer to Granite Vision (Team et al., 2025). For the Self-curated Data, we use PaddleOCR as our OCR model. We first filter arXiv papers to include only those published in venues

| Model | Abstract | Author | Caption | Equation | Figure | Footer |
|---|---|---|---|---|---|---|
| DinoV2 | 86.4 | 9.0 | 45.2 | 14.9 | 90.5 | 3.6 |
| MAE | 84.1 | 17.9 | 20.7 | 33.4 | 90.8 | 23.8 |
| Dolphin | 73.8 | 21.6 | 48.1 | 25.7 | 91.5 | 32.9 |
| Pix2Struct | 80.6 | 26.3 | 21.6 | 15.3 | 83.1 | 27.7 |
| AIMv2 | 85.8 | 5.5 | 27.7 | 15.5 | 90.4 | 19.4 |
| SigLIP 2 | 85.9 | 23.1 | 52.6 | 34.7 | 91.6 | 31.6 |
| DAVE | 87.4 | 35.3 | 54.1 | 53 | 91.5 | 17.9 |

| Model | List | Paragraph | Reference | Section | Table | Title |
|---|---|---|---|---|---|---|
| DinoV2 | 42.2 | 10.3 | 86.1 | 3.2 | 68.5 | 0.0 |
| MAE | 40.8 | 10.2 | 88.8 | 9.2 | 88.1 | 26.5 |
| Dolphin | 68.7 | 11.5 | 88.0 | 9.7 | 84.3 | 50.2 |
| Pix2Struct | 33.9 | 13.7 | 87.3 | 5.4 | 62.2 | 64.4 |
| AIMv2 | 45.8 | 9.8 | 86.2 | 2.8 | 80.3 | 39.0 |
| SigLIP 2 | 51.8 | 13.2 | 88.7 | 7.5 | 90.9 | 48.8 |
| DAVE | 66.4 | 20.6 | 91.6 | 27 | 87.9 | 49.7 |

Table 9: Evaluation results on DocBank by category.

or journals, ensuring a high-quality source corpus. During post-processing, we remove all bounding boxes with confidence scores below 0.5 to ensure accurate OCR localization. We then randomly resize, crop, and add noise to prevent the model from overfitting to a specific PDF size. Next, we normalize all bounding box coordinates to the range [0, 999] with respect to the new size. To control task difficulty, we discard bounding boxes that contain either too few words (e.g., single tokens) or excessively long paragraphs. Finally, from each remaining (category, bounding box, text) tuple, we construct diverse training examples for bbox-to-text, text-to-bbox, and bbox-to-category tasks. The data composition is shown in Table 8.

### B.3 Weight-Merging

For the weight-merging, we randomly sample 10k document and web images from self-supervised pretraining as our training data. We use the AdamW Optimizer (Loshchilov & Hutter, 2017), a learning rate of 1e-4, and a batch size of 32 to train for 20 epoches. The only trainable component is the newly introduced merge coefficient, while all the vision encoders remain frozen.

## C Classic Document and web Tasks

### C.1 DocBank

DocBank is a large-scale benchmark for document layout analysis, providing token-level annotations that capture both textual and structural information. It is widely used to evaluate models on tasks such as text detection, segmentation, and understanding of document formatting. In our work, we formulate DocBank as Document element recognition, where the task is the predict the bounding box of a specific element given a document image.

**Implementation**. DocBank consists of 13 categories. For each category, we finetune the vision encoder with an attention pooling network on the corresponding training set, and perform evaluation on the validation set. Before training, each image is processed and resized according to the default preprocessor for the specific vision encoder. For each category, we train with 5 epoch, a learning rate of 1e-5, weight decay of 1e-4, and a batch size of 32. We use Smooth L1 objective (Huber, 1964) for the training. The per-category result is shown in Table 9.

### C.2 DocLayNet

DocLayNet is a large-scale dataset for document layout analysis that contains richly annotated page images collected from diverse sources such as scientific articles, reports, and business documents. It provides pixel-level segmentation masks for a broad set of layout elements, enabling fine-grained

evaluation of models on tasks like detection, segmentation, and structural understanding. In our work, we use Doclaynet to perform semantic segmentation.

**Implementation**. Doclaynet consists of 11 distinct classes, and in our implementation, we introduce an additional class for the area with no class label. We finetune the vision encoder along with an MLP projector to project the visual feature patches into a segmentation mask. Then we use crossentropy (Shannon, 1948) between the prediction and the ground truth segmentation as objective and train on the training set. We train for 2 epoch, with a learning rate of 1e-5, weight decay of 1e-4, and a batch size of 32.

**Ablation on the generalist encoder**. In this section, we conduct an ablation on the improvement to RICO-SCA when we incorporate the generalist encoder feature. As shown in Table 10, it is clear tha the generalist encoder provide high-level semnatic features that are important for classification tasks.

| Model | RICO-SCA |
|---|---|
| SigLIP 2 | 93.3 |
| DAVE-specialized | 90.7 |
| DAVE | 92.3 |

Table 10: Performance on the RICO-SCA benchmark.

### C.3 RICO-SCA

RICO-SCA is a benchmark derived from the RICO dataset, which contains large-scale mobile application user interfaces. The SCA (Semantic Component Annotation) extension enriches each screen with detailed semantic labels for UI elements such as class type, buttons, images, text fields, and navigation components. In our work, we use RICO-SCA to evaluate on web UI classification.

**Implementation**. We finetune the vision encoder together with an attention pooling network to classify the class type of a given web UI image, using the provided training set. We train for 3 epoch, a learning rate of 1e-5, a weight decay of 1e-4, and a batch size of 32. We then evaluate the finetuned model on the validation set.

## D VLM AS EVALUATOR

| Dataset | #Image | Description |
|---|---|---|
| AI2D | 2k | Chart understanding |
| DocVQA | 10k | Document understanding |
| InfoVQA | 2k | Infographic reasoning |
| ChartQA | 17k | Chart understanding |
| Pixmo-Doc | 250k | Document understanding |
| MultiUI | 2m | Web localization and understanding |
| LLaVA 665K | 340k | General visual instruction data |

Table 11: Dataset description for VLM training.

Here we provide the detail implementation and data composition on using VLMs as evaluator for vision encoders.

**Data Composition**. We use a set of instruction tuning data with a focus on document and web understanding for the VLM training. We then use the Mind2Web training set to finetune our VLM on web agentic tasks. The full data composition can be found in Table 11

**Implementation**. The training is conducted with 32 H200 GPUs. We use the same architecture as described in Section B. For image tilting, we use tilting to split the image into size of 336x336 for AIMv2, and 384x384 for other vision encoders. The instruction tuning hyperparameter and finetuning is the same as in 7, except that Mind2Web is finetuned for 2 epoch.

## D.1 EVALUATION STANDARD ERRORS

| Model | AI2D | ChartQA | DocVQA | InfoVQA | RealWorldQA | TextVQA |
|-------|------|---------|--------|---------|-------------|---------|
| llama-Siglip | 0.88 | 0.99 | 0.55 | 0.85 | 1.80 | 0.64 |
| llama-DAVE | 0.88 | 0.97 | 0.47 | 0.87 | 1.77 | 0.62 |
| qwen-Siglip | 0.79 | 0.85 | 0.36 | 0.74 | 1.70 | 0.58 |
| qwen-DAVE | 0.78 | 0.76 | 0.35 | 0.76 | 1.80 | 0.59 |

Table 12: Standard errors in vision-language benchmarks.

In this section, we provide the standard errors for most vision-language benchmarks. As shown in Table 12, the standard errors remain are reasonable, indicating that the model's performance is stable and consistent across benchmarks.

## D.2 EVALUATION ON CROSS-DOMAIN TASKS

| Model | CMMMU | DTCBench |
|-------|-------|----------|
| SigLIP 2 | 21.4 | 29.2 |
| DAVE | 26.1 | 38.3 |

Table 13: Evaluation results on CMMMU and DTCBench.

Here, we provide results on DTCBench for Korean document understanding and CMMMU for Chinese visual question answering. In Table 13, DAVE's consistent improvements on these benchmarks show that it can generalist to cross domain

# E ABLATIONS

In this section, we provide the implementation and evaluation details of our ablation studies, along with additional ablations on DAVE.

## E.1 INTER-PATCH VARIANCE

We randomly sample 1000 document and web images from our pretraining data to conduct the analysis. For each image, we divide it into non-overlapping $16 \times 16$ patches and compute the per-patch standard deviation of pixel intensities after normalization. This provides a distribution of inter-patch variance that reflects the inherent local variability in different domains. As discussed in Section 5.4, document-like images exhibit consistently lower variance across patches, while web-like and natural images demonstrate a broader spread. These differences highlight the difference in low-level representation, particularly structural and spatial, between natural and document/web images.

## E.2 MULTI-ENCODER SETUP

Given a setup consisting of SigLIP 2 and a specialized encoder, the images are first processed separately by their respective preprocessor. The processed images are then passed into the corresponding vision encoders to obtain visual patch features. Finally, the patch features are concatenated channel-wise. In scenarios where the specialized vision encoder produces a different number of patch features compared to SigLIP 2, we apply linear interpolation on the specialized features to match the patch count.

## E.3 GENERALIZATION TO UNSEEN GENERALIST ENCODER

We denote the specialized encoder within DAVE as DAVE-spec. To evaluate whether DAVE-spec can adopt to unseen generalist encoder, we pair it with with AIMv2 during instruction tuning. The

| Model | Doc | Web |
|---|---|---|
| DAVE | 63.4 | 68.2 |
| DAVE-spec + AIMv2 | 64.5 | 64.8 |

Table 14: Evaluation results on document and web understanding.

result is show in Table 14. The comparable performance with DAVE highlights the generalizability of DAVE-spec to novel vision-language and agent frameworks.

### E.4 EFFECTS OF FUSING SIGLIP 2 FEATURES

In Table 15, we ablate the effect of fusing SigLIP 2 on high-level semantic tasks.

| Model | RICO-SCA |
|---|---|
| SigLIP 2 | 93.3 |
| DAVE-specialized | 90.7 |
| DAVE | 92.8 |

Table 15: Evaluation results on RICO-SCA.

