# OpenReview forum: "DAVE: A VLM Vision Encoder for Document Understanding and Web Agents"
_ICLR.cc/2026/Conference — ICLR 2026 Poster_

### Official Review · Reviewer_zSTT · 2025-10-26

**Soundness:** 2
**Presentation:** 2
**Contribution:** 2
**Rating:** 4
**Confidence:** 5

**Summary:**

This paper introduces DAVE, a specialized vision encoder for VLMs targeting document understanding and web agent tasks. The authors propose a two-stage training approach: (1) self-supervised pretraining using MAE on unlabeled document and web images, and (2) supervised autoregressive pretraining with limited annotated data. Key innovations include a weight-merging scheme to create decoder-agnostic encoders and ensemble training to combine general visual features with domain-specific representations. Experiments demonstrate improvements over existing encoders on document, web, and agent benchmarks.

**Strengths:**

The focus on document and web understanding addresses a real limitation of current VLMs.
Testing across multiple benchmarks (document tasks, VQA, web agent tasks) demonstrates broad applicability.

**Weaknesses:**

The paper lacks deeper analysis of why this specific combination of techniques works. Why is ensemble training with SigLIP2 optimal? What are the theoretical guarantees of the weight merging approach?

While the combination is new, individual components are well-established, such as MIM pretraining and Visual-language alignment.

**Questions:**

1. How sensitive is the approach to the choice of generalist encoder? Would similar improvements be seen with other encoders?

2. More visual encoders should be compared, such as [1][2][3].
[1] Web-DINO: Scaling Visual Representation Learning
[2] A Token-level Text Image Foundation Model for Document Understanding
[3] RADIOv2.5: Improved Baselines for Agglomerative Vision Foundation Models

3. A similar idea to Vary, which uses two encoders to represent the text scenes.

---

> ### Author Response · Authors · 2025-11-21
> **Response to reviewer zSTT (part 1/2)**
>
> Dear reviewer zSTT, thank you for your insightful and thorough comments. We are very happy that you found our method widely applicable to document and web tasks. We address your remaining questions as below:
> ***
> **Q1:** The paper lacks deeper analysis of why this specific combination of techniques works.
>
> **A1:** Thank you for the constructive concern. We agree that providing deeper analysis strengthens the paper. In the revised version, we include a more detailed examination of why this specific combination of techniques is effective. In the self-supervised stage, the model learns rich spatial and structural representations of documents, which is reflected in the improvements shown in Table 3. In the supervised stage, the model is further aligned with downstream tasks such as grounding and recognition. We also apply model merging so that the resulting visual representation becomes decoder-agnostic, enabling the LLM to use it effectively. A more comprehensive ablation study of design choices can be found in Figure 2.
>
> **Q1 continue:** Why is ensemble training with SigLIP2 optimal?
>
> **A1 continue:** We would like to note that we never claimed that SigLIP is optimal, and we chose SigLIP 2 for its implementation simplicity and its wide adoption in many existing VLMs literature. In addition, we provide results using AIMv2 as the generalist encoder and achieve similar improvements. As shown in the table, DAVE with AIMv2 performs comparably with SigLIP 2, which indicates that our method is not sensitive to the choice of generalist encoder.
> | Model               | ai2d | OCR  | Doc  | Info | Chart | Visual | SV2  | Websrc  |
> |---------------------|------|------|------|------|--------|--------|------|------|
> | DAVE - with Siglip 2 | 59.6 | 62.2 | 82.1 | 50.2 | 63.1   | 59.2   | 64.5 | 82.6 |
> | DAVE - with aimv2    | 61.2 | 62.0 | 82.1 | 50.3 | 66.9   | 56.0   | 63.8 | 82.1 |
>
> ***
> **Q1 continue:** What are the theoretical guarantees of the weight merging approach?
>
> **A1 continue:** The theoretical guarantees of weight merging follow from interpreting each trained model as defining a posterior distribution over parameters [1]. If these posteriors are isotropic Gaussians centered at the model weights, then averaging the parameters is the exact solution that maximizes the joint posterior. Fisher Merge extends this by adopting the Laplace approximation, under which each posterior is a Gaussian whose precision is given by the Fisher information; in this setting, the Fisher-weighted average is the optimal solution.
> Our gradient-based approach does not rely on these specific posterior assumptions and is an approximation to the optimal solution. If the true model posterior is isotropic or Laplace-Gaussian, then our gradient-based approach can be at most as good as the optimal solutions. But when the true posterior deviates these distributions, our gradient-based approach could provide a more accurate approximation. The superior performance of our method in Table 4a suggests that this is the case for merging our vision encoders.
>
> [1] Matena, Michael S., and Colin A. Raffel. "Merging models with fisher-weighted averaging." NeurIPS 2022.
> ***
> **Q2:** While the combination is new, individual components are well-established, such as MIM pretraining and Visual-language alignment.
>
> **A2:** We emphasize that the novelty of our approach lies in the design of our training framework, not the individual components we apply. For example, we innovate by decoupling self-supervised and supervised objectives based on the constraints of documents and web screenshots data. This approach differs from vision encoders that are pretrained jointly with self-supervised and supervised objectives.  Concretely, the staged combination of self-supervised learning, autoregressive supervised training, and model merging allows us to efficiently leverage large-scale unlabeled data alongside limited annotated examples.
> ***

---

> ### Author Response · Authors · 2025-11-21
> **Response to reviewer zSTT (part 2/2)**
>
> **Q3:** How sensitive is the approach to the choice of generalist encoder? Would similar improvements be seen with other encoders?
>
> **A3:** Our approach is not sensitive to the choice of generalist encoder, as a different generalist encoder, AIMv2, achieves similar improvements compared to SigLIP 2 (See Q 1).
> ***
> **Q4:** More visual encoders should be compared, such asWeb-DINO: Scaling Visual Representation Learning, A Token-level Text Image Foundation Model for Document Understanding, RADIOv2.5: Improved Baselines for Agglomerative Vision Foundation Models.
>
> **A4:** As suggested, we included the additional baselines in our comparison. As shown in the table below, the results demonstrate that DAVE consistently achieves better performance on document and web understanding tasks.
> | Model     | ai2d | OCR  | Doc  | Info | Chart | Visual | SV2  | Websrc  |
> |-----------|------|------|------|------|--------|--------|------|------|
> | Web-DINO  | 54.6 | 20.3 | 35.8 | 30.1 | 33.7   | 30.0   | 54.3 | 40.5 |
> | RADIOv2.5 | 55.0 | 49.0 | 74.4 | 37.1 | 45.1   | 50.3   | 46.9 | 70.2 |
> | TokenFD   | 58.7 | 67.5 | 80.8 | 48.0 | 65.9   | 57.7   | 60.0 | 76.4 |
> | DAVE      | 59.6 | 62.2 | 82.1 | 50.2 | 63.1   | 59.2   | 64.5 | 82.6 |
> ***
> **Q5:** A similar idea to Vary, which uses two encoders to represent the text scenes.
>
> **A5:** We appreciate the reference to Vary and will include it in our revision. However, we note that our methodology differs significantly. First, regarding architecture, Vary performs fusion only after training the specialized encoder. In contrast, we integrate generalist features during pretraining, allowing our encoder to learn complementary representations. Second, regarding objectives, Vary relies on supervised learning, whereas we identify self-supervised learning as critical for spatial tasks. Table 3 validates this, showing that SSL-trained encoders consistently outperform those using only language objectives. Furthermore, unlike Vary, which is tightly coupled to its decoder, our weight-merging scheme ensures our encoder remains robust and generalizable to diverse LLMs. Finally, our data scope is broader: while Vary targets document vocabulary, we cover both documents and web screenshots with a specific focus on structural grounding.

---

> ### Author Response · Authors · 2025-11-26
> **Follow up on our response**
>
> Dear Reviewer zSTT,
>
> We would like to kindly follow up on our previous response and check whether there is anything else we could help clarify. We would be glad to provide any additional information or details that you may find helpful. Thank you again for your thoughtful and constructive review!
>
> Best regards,
>
> Authors

---

### Official Review · Reviewer_Q3WH · 2025-11-01

**Soundness:** 2
**Presentation:** 2
**Contribution:** 2
**Rating:** 4
**Confidence:** 3

**Summary:**

The paper presents DAVE, a vision encoder tailored for document understanding and web agent tasks within vision-language models. It introduces a two-stage pretraining framework combining self-supervised masked autoencoding on large unlabeled datasets with a supervised autoregressive stage for tasks such as OCR, layout prediction, and web grounding. To enhance generalization across architectures, the authors employ a weight-space merging scheme that combines encoders trained with different text decoders, producing a decoder-agnostic encoder, and an ensemble training strategy that fuses general semantic features from SigLIP2 with DAVE’s specialized structural features.

**Strengths:**

* The paper is clearly written and easy to follow.
* The research question of vision encoder for document images is important.

**Weaknesses:**

* The contribution of the paper is limited. The training methods employed in the paper including self-supervised pre-training, model merging are well-established ideas such as MAE self-supervised learning [1], model soups [2].

* Although the experiments encompass a wide range of benchmarks (e.g., DocVQA, ChartQA, Mind2Web), the paper provides limited analysis of cross-domain generalization—such as performance on non-English documents—as well as robustness and interpretability aspects.

[1] Masked Autoencoders Are Scalable Vision Learners.

[2] Model soups: averaging weights of multiple fine-tuned models improves accuracy without increasing inference time.

**Questions:**

There are a few typos in the paper:
* Line 240: "Appendix X"
* Line 268: It should be "Qwen2.5-7B-Instruct."

---

> ### Author Response · Authors · 2025-11-21
> **Response to reviewer Q3WH (part 1/1)**
>
> Dear reviewer Q3WH, we thank you for your thoughtful comments. We are glad that you appreciate the importance of a vision encoder for document images. We response the remaining questions as follow:
> ***
> **Q1:** The contribution of the paper is limited. The training methods employed in the paper including self-supervised pre-training, model merging are well-established ideas such as MAE self-supervised learning [1], model soups [2].
>
> **A1:** We emphasize that the novelty of our approach lies in the training framework design rather than the individual objectives. Existing vision encoder pretraining like SigLIP 2 incorporates both self-supervised and supervised methods to learn rich representation, and we innovate by decoupling these objectives based on the unique constraints of documents and web screenshots. Concretely, the staged combination of self-supervised learning, autoregressive training, and model merging allows us to efficiently leverage large-scale unlabeled data alongside limited annotated examples. This differs substantially from traditional vision encoders that apply either one objective or multiple objectives jointly.
> Regarding Model Soup, our application diverges from the standard paradigm. While prior works typically use merging to combine specialized task experts, we introduce a novel decoder-diversity merging scheme. We merge encoders trained with different text decoders to prevent the vision encoder from overfitting to a specific language model. This specific application—creating a decoder-agnostic encoder via merging—is a novel contribution to the VLM design space.
> ***
> **Q2:** Although the experiments encompass a wide range of benchmarks (e.g., DocVQA, ChartQA, Mind2Web), the paper provides limited analysis of cross-domain generalization—such as performance on non-English documents—as well as robustness and interpretability aspects.
>
> **A2:** Thank you for your concerns on cross-domain generalization. In the table below, we provide CMMMU for Chinese general visual understanding and DTCBench for Korean document understanding. We can observe that DAVE improves non-english document understanding while maintaining performance on non-english general VQAs compared to SigLIP 2, highlighting its cross-domain generalization.
> | Model    | CMMMU | DTCBench |
> |----------|-------|----------|
> | SigLIP 2 | 21.4  | 29.2     |
> | DAVE     | 26.1  | 38.3     |
>
> Regarding robustness and interpretability, our model offers distinct advantages over existing vision encoders. First, our weight-merging scheme ensures the model is significantly more robust to the choice of LLM decoder, preventing the overfitting common in standard encoders. Second, our ensemble architecture enhances interpretability—specifically regarding feature attribution. By comparing our ensemble against the standalone generalist encoder, we can explicitly isolate and attribute performance gains in document and web tasks to the additional structural and spatial features provided by our specialized encoder.
> ***
> **Q3:** Typos in the paper
>
> **A3:** Thank you! We have fixed them in the revision and are actively searching for more typos.
> ***

---

> ### Author Response · Authors · 2025-11-26
> **Follow up on our response**
>
> Dear Reviewer Q3WH,
>
> We would like to kindly follow up on our previous response and check whether there is anything else we could help clarify. We would be glad to provide any additional information or details that you may find helpful. Thank you again for your thoughtful and constructive review!
>
> Best regards,
>
> Authors

---

### Official Review · Reviewer_rHNd · 2025-11-02

**Soundness:** 2
**Presentation:** 2
**Contribution:** 2
**Rating:** 4
**Confidence:** 5

**Summary:**

DAVE presents a compelling approach to developing a specialized vision encoder for document and web understanding, demonstrating strong empirical results. The innovative MAE reconstruction and decoder-agnostic merging scheme are notable contributions. However, the paper suffers from several weaknesses, primarily in providing insufficient experimental evidence or logical clarity to fully support certain design choices and claims. Addressing these points, particularly through more rigorous ablation studies and transparent comparative methodologies, would significantly strengthen the paper's impact and credibility.

**Strengths:**

1. The paper proposes a modification to the standard MAE objective by reconstructing raw pixel values directly, rather than normalized pixel values. This aims to stabilize training, particularly for document and web images which exhibit low inter-patch variance. While further empirical evidence directly linking this change to stability would strengthen the claim, the approach itself is a thoughtful adaptation to domain-specific data characteristics.
2.  A significant strength is the introduction of a distillation-based model merging scheme. This approach trains multiple encoder instances, each aligned with a different text decoder, and then learns optimal coefficients to combine their weights into a single, decoder-agnostic encoder. This method effectively addresses the challenge of encoder overfitting to specific decoders and enhances compatibility across diverse VLM architectures.
3.  The authors leverage a vast amount of data for both self-supervised and supervised pretraining, including large-scale unlabeled document/web images and curated annotated datasets. This extensive data utilization, combined with the proposed training strategies, leads to DAVE achieving impressive performance across a wide range of document, general VQA, and web agent benchmarks, often outperforming strong baselines like SigLIP 2.

**Weaknesses:**

1.  The core hypoththesis of this paper is that current VLM vision encoders "lack the robust structural and spatial information essential for document understanding and web agents." However, this assertion is made without references or experimental evidence. Given the existence of numerous VLMs works for document understanding, a more detailed explanation is needed to clarify why these models specifically lack the required powerful structural and spatial information. The observed performance gains of DAVE, while compelling, could also be attributed to its extensive and specialized training data rather than a fundamental architectural superiority in capturing these specific features, necessitating a more rigorous analysis.
2.  For document and web images, the authors argue that low inter-patch variance leads to training instability, and therefore they reconstruct from raw pixel values rather than normalized pixel values. However, the authors only show a difference in inter-patch variance distribution between ImageNet-like, document-like, and web-like images, without providing experimental evidence which directly demonstrating that this variance difference leads to training instability with the standard MAE reconstruction method.
3.  DAVE combines a specialized encoder ($\phi_{spec}$) for low-level structural features with a generalist encoder ($\phi_{gen}$) for high-level semantics. However, if we  fine-tuning a strong, generalist encoder like SigLIP 2 (with DAVE's training protocol and data) would yield comparable results?
4.   When comparing DAVE with other models like SigLIP 2, it's unclear whether the original decoder in SigLIP2 was replaced by the same LLM  decoder used in DAVE?
5. In Table 3, DAVE’s performance on RICO-SCA is slightly lower than that of SigLIP 2. The authors explain this by stating that the task emphasizes semantic understanding, which DAVE was not designed to tackle. However, this explanation directly contradicts the motivation claimed by the authors for integrating SigLIP 2 into DAVE, namely to provide high-level semantic capabilities for better document understanding. This inconsistency needs to be addressed.
6. Figure 2, which illustrates the incremental impact of design choices, uses terms like "Scratch Decoder" and "Pretrained Decoder" without clear definitions. Furthermore, some data points in Figure 2 can not be found in the tables in the paper, making it difficult for readers to fully understand and verify the ablation results.
7. In Table 4c, when comparing DAVE with multi-encoder setups (e.g., SigLIP 2 + DiT, PS, DP), it is not explicitly stated whether these models underwent the same extensive training pipeline as DAVE — including self-supervised pretraining, multi-decoder supervised training, and model merging. If these pre-trained models were merely concatenated and directly evaluated on the benchmarks without following the same training procedure, the comparison would be unfair.
8. The comparison in Table 4d between DAVE and a fine-tuned SigLIP 2 is potentially unfair. The fine-tuned SigLIP 2 is described as being trained "only on the supervised data," implying it did not undergo DAVE's full two-stage pipeline, including self-supervised pretraining and multi-decoder merging.

**Questions:**

see Weaknesses.

---

> ### Author Response · Authors · 2025-11-21
> **Response to reviewer rHNd (part 1/3)**
>
> Dear reviewer rHND, We sincerely thank Reviewer rHND for the insightful feedback. We appreciate the recognition that our approach to leveraging both unlabeled and labeled data is significant. We address the remaining questions below
> ***
> **Q1:** The core hypothesis of this paper is that current VLM vision encoders "lack the robust structural and spatial information essential for document understanding and web agents." However, this assertion is made without references or experimental evidence.
>
> **A1:** To measure spatial and structural information, we rely on dense prediction tasks, which are widely used to evaluate the quality of spatial and structural representations [1,2]. As shown in Table 3, DAVE outperforms other encoders on document segmentation and recognition benchmarks, indicating that it captures spatial and structural information more effectively than existing VLM encoders. Notably, specialized encoders such as Dolphin, despite being trained exclusively on annotated image–text pairs, perform significantly worse than models trained with self-supervised objectives. This gap highlights the importance of self-supervised learning for developing robust spatial understanding, consistent with prior work [1,2]. Building on these works, we trained a document-specialized encoder using self-supervised objectives, further improving spatial and structural representation for documents.
>
> **Q1 continue:** Given the existence of numerous VLMs works for document understanding, a more detailed explanation is needed to clarify why these models specifically lack the required powerful structural and spatial information.
>
> **A1 continue:** We clarify that our work focuses specifically on the vision encoder component, rather than the VLM system as a whole. While several VLMs for document understanding do exist [3,4,5], they typically rely on generalist encoders (e.g., CLIP) or encoders trained on limited annotated data. Consequently, these VLMs inherit the spatial and structural deficiencies discussed above. Our work distinguishes itself by proposing a novel vision encoder designed to be seamlessly integrated into a wide range of VLMs.
>
> **Q1 continue:** The observed performance gains of DAVE, while compelling, could also be attributed to its extensive and specialized training data rather than a fundamental architectural superiority in capturing these specific features, necessitating a more rigorous analysis.
>
> **A1 continue:** We agree that domain-specific data is crucial. To demonstrate that our performance gains stem from our training methodology rather than raw data scale, we benchmark against Dolphin, a recent-state-of-the-art. As shown in Tables 1 and 3, Dolphin was trained on 30 million annotated samples—a significantly larger labeled corpus than DAVE's. The fact that DAVE outperforms Dolphin on both VQA and dense prediction tasks, despite using fewer annotated samples, validates our core contribution: the effectiveness of decoupling self-supervised learning to efficiently leverage unlabeled data.
> ***
>
> **Q2:** For document and web images, the authors argue that low inter-patch variance leads to training instability, and therefore they reconstruct from raw pixel values rather than normalized pixel values. However, the authors only show a difference in inter-patch variance distribution between ImageNet-like, document-like, and web-like images, without providing experimental evidence which directly demonstrating that this variance difference leads to training instability with the standard MAE reconstruction method.
>
> **A2:** Following the reviewer's suggestion, we have updated the appendix (Figure 4) with a loss curve illustrating the gradient explosion and training divergence caused by normalized pixel values.
> ***
> [1] Maninis, Kevis-Kokitsi, Kaifeng Chen, Soham Ghosh, Arjun Karpur, Koert Chen, Ye Xia, Bingyi Cao et al. "TIPS: Text-image pretraining with spatial awareness." ICLR 2025
>
> [2] Naeem, Muhammad Ferjad, Yongqin Xian, Xiaohua Zhai, Lukas Hoyer, Luc Van Gool, and Federico Tombari. "Silc: Improving vision language pretraining with self-distillation." ECCV 2024
>
> [3] Hu, Anwen, Haiyang Xu, Jiabo Ye, Ming Yan, Liang Zhang, Bo Zhang, Ji Zhang, Qin Jin, Fei Huang, and Jingren Zhou. "mplug-docowl 1.5: Unified structure learning for ocr-free document understanding." EMNLP 2024
>
> [4] Kim, Geewook, Teakgyu Hong, Moonbin Yim, JeongYeon Nam, Jinyoung Park, Jinyeong Yim, Wonseok Hwang, Sangdoo Yun, Dongyoon Han, and Seunghyun Park. "Ocr-free document understanding transformer." ECCV 2022
>
> [5] Liu, Yuliang, Biao Yang, Qiang Liu, Zhang Li, Zhiyin Ma, Shuo Zhang, and Xiang Bai. "Textmonkey: An ocr-free large multimodal model for understanding document." arXiv
> ***

---

> ### Author Response · Authors · 2025-11-21
> **Response to reviewer rHNd (part 2/3)**
>
> **Q3:** DAVE combines a specialized encoder for low-level structural features with a generalist encoder for high-level semantics. However, if we fine-tuning a strong, generalist encoder like SigLIP 2 (with DAVE's training protocol and data) would yield comparable results?
>
> **A3:** As shown in Table 4d, DAVE outperforms SigLIP 2 finetuned on the annotated data. Also see Q8 for further clarification.
> ***
>
> **Q4:** When comparing DAVE with other models like SigLIP 2, it's unclear whether the original decoder in SigLIP2 was replaced by the same LLM decoder used in DAVE?
>
> **A4:** We clarify that for all baselines—whether they are encoder-only (like SigLIP 2) or encoder-decoder models (like AIMv2, Dolphin, and Pix2Struct)—we utilized only their vision encoder component. We then performed instruction tuning by pairing every vision encoder with the exact same LLM decoder.
> ***
> **Q5:** In Table 3, DAVE’s performance on RICO-SCA is slightly lower than that of SigLIP 2. The authors explain this by stating that the task emphasizes semantic understanding, which DAVE was not designed to tackle. However, this explanation directly contradicts the motivation claimed by the authors for integrating SigLIP 2 into DAVE, namely to provide high-level semantic capabilities for better document understanding. This inconsistency needs to be addressed.
>
> **A5:** This is a very interesting observation on the performance of DAVE on RICO-SCA. Theoretically, DAVE should perform at least as well as SigLIP 2 because it includes features from SigLIP 2.
> We present the results of testing on RICO-SCA using only the specialized encoder model in DAVE. As shown, incorporating SigLIP 2 improves performance, supporting our claim that SigLIP 2 provides high-level semantic information. We attribute DAVE’s suboptimal performance compared to SigLIP 2 to the increased hidden dimensionality. Since DAVE’s embedding dimension is twice as large as that of SigLIP 2, it is more difficult to pool and process the information with the same one-layer MLP prediction head.
> | Model             | RICO-SCA |
> |-------------------|----------|
> | SigLIP 2          | 93.3     |
> | DAVE-specialized  | 90.7     |
> | DAVE              | 92.8     |
> ***

---

> ### Author Response · Authors · 2025-11-21
> **Response to reviewer rHNd (part 3/3)**
>
> **Q6:** Figure 2, which illustrates the incremental impact of design choices, uses terms like "Scratch Decoder" and "Pretrained Decoder" without clear definitions. Furthermore, some data points in Figure 2 can not be found in the tables in the paper, making it difficult for readers to fully understand and verify the ablation results.
>
> **A6:** We appreciate the feedback regarding the clarity of our figure. To clarify, 'Scratch Decoder' implies that the text decoder is initialized randomly during supervised pretraining, whereas 'Pretrained Decoder' utilizes a pretrained LLM checkpoint. 'Ensemble' indicates the incorporation of frozen features from a generalist encoder. We have included a table below comparing these settings against a SigLIP 2 baseline for better visualization, and we will revise the final manuscript to make these definitions explicit.
> ***
> **Q7:** In Table 4c, when comparing DAVE with multi-encoder setups (e.g., SigLIP 2 + DiT, PS, DP), it is not explicitly stated whether these models underwent the same extensive training pipeline as DAVE — including self-supervised pretraining, multi-decoder supervised training, and model merging. If these pre-trained models were merely concatenated and directly evaluated on the benchmarks without following the same training procedure, the comparison would be unfair.
>
> **A7:** We would like to first point out that applying our full training pipeline for each multi-encoder configuration is equivalent to initializing the specialized encoder with one of the pretrained encoders prior to the self-supervised training. We have tried initializing with different encoders, including SigLIP 2, MAE-Web, and Dolphin, but did not observe significant gains. Therefore, we opted for the simplest setting and trained the specialized encoder from scratch.
>
> We note that the multi-encoder configurations did not undergo the supervised and model merging training stages, as those specialized encoders were already pretrained on document data. Nevertheless, to demonstrate the effectiveness of applying our supervised and model merging training stages on other multi-encoder setup, we selected SigLIP 2 + DiT—the best-performing combination—and reported SigLIP 2 + DiT – Finetune on these stages. While this fine tuning significantly improves performance, DAVE still achieves the best results.
> | Model                     | ai2d | OCR  | Doc  | Info | Chart | Visual | SV2  | Web  |
> |---------------------------|------|------|------|------|-------|--------|------|------|
> | SigLIP 2 + DiT            | 49.9 | 49.5 | 60.1 | 41.1 | 50.6  | 42.8   | 41.1 | 58.8 |
> | SigLIP 2 + DiT – Finetune | 55.7 | 55.5 | 70.1 | 41.1 | 55.8  | 48.2   | 46.0 | 64.8 |
> | DAVE                      | 59.6 | 62.2 | 82.1 | 50.2 | 63.1  | 59.2   | 64.5 | 82.6 |
> ***
> **Q8:** The comparison in Table 4d between DAVE and a fine-tuned SigLIP 2 is potentially unfair. The fine-tuned SigLIP 2 is described as being trained "only on the supervised data," implying it did not undergo DAVE's full two-stage pipeline, including self-supervised pretraining and multi-decoder merging.
>
> **A8:** We clarify that the 'SigLIP-finetune' baseline undergoes both the supervised training stage and the model merging stage, not just supervised training. As noted in our response to the previous question, applying the full pipeline (starting with self-supervised learning) did not yield performance gains for vision encoder baselines. Consequently, we finetuned SigLIP 2 using only the supervised and model merging stages, as this configuration yielded the best performance.
> ***

---

> ### Author Response · Authors · 2025-11-25
>
> Dear Reviewer rHNd, we would like to thank you once again for your review, and for improving your evaluation of our paper in response to our rebuttal.

---

### Official Review · Reviewer_gMFC · 2025-11-10

**Soundness:** 3
**Presentation:** 3
**Contribution:** 3
**Rating:** 8
**Confidence:** 4

**Summary:**

The main contribution is a vision encoder architecture for document and web understanding evaluated on a broad array of benchmarks and reports improvements over strong baselines. The use of PlotQA, ChartQA, FinTabNet is well motivated. Key contribution is enabling data efficient training in specialized domains.
Compared to SigLIP2 and AIMv2 in particular, the submission shows that
1) On Document Understanding Benchmarks (OCRBenchm DocVQA, InfoVQA) the gains are meaningful and suggests that the proposed architecture provides a non trivial improvement in the document domain
2) On General VQA the results are a bit mixed which calls for more understanding of the regressions esp wrt to SigLiP2

Overall in terms of novelty and significance, compared to known encoder types and training strategies (contrastive + self-supervised for SigLIP2l autoregressive multimodal for AIMv2) the core contribution lies in demonstrating broad generalized across heterogenous benchmarks.

**Strengths:**

The breadth of evaluations and ablations is a key strength.
[1] Thorough benchmarking on Document understanding, general VQA, and Web understanding benchmarks: We see evaluation across a wide variety of benchmarks and baseline models. Using both Qwen-2.5-7B-Instruct and Llama-3.2-3B-Instruct decoders further solidify the alignment on different text decoders
[2] Very strong improvements on Mind2Web (Table 2)
[3] Clean ablation setups are useful to justify the training setup. Table 4 c,d are esp interesting contribution comparing different multi encoder settings

**Weaknesses:**

[1] average perfomance is used across all results which might under or over estimate the performance. Having some best of N results using some confidence or majority voting would further help understand the variance of the model performance better.
[2] there might be potential domain bias, example overfitting to synthetic chart styles in PlotQA or financial domain layouts in FinTabNet. Might have been useful to detect and discuss such biases.

**Questions:**

[1] We need to have confidence intervals for the results esp when comparing SigLIP2 to DAVE using Qwen-2.5-7B-Instruct model in Table 1.
[2] A little more discussion of regression on RealWorldQA and TextVQA would be useful to understand if regressions on general VQA capabilities are real
[3] we processed 500K PDF samples from arXiv with an OCR model (Cui et al., 2025), formulating them into additional recognition
and grounding tasks. What kinds of post processing was done on the data to ensure high quality examples?

---

> ### Author Response · Authors · 2025-11-21
> **Response to reviewer gMFC (part 1/1)**
>
> We thank Reviewer gMFC for the time and valuable feedback. We appreciate the positive assessment of our work, particularly the recognition of our evaluation's breadth. We address the remaining questions below.
> ***
> **Q1:** average performance is used across all results which might under or over estimate the performance. Having some best of N results using some confidence or majority voting would further help understand the variance of the model performance better.
>
> **A1:** We thank the reviewer for the insightful suggestion regarding model variance. For our evaluation, we use the `lmms-eval` framework to ensure that the results are fully reproducible, with standard prompts and deterministic hyperparameters. While this framework does not currently support inference-time strategies like 'Best-of-N' or majority voting for these benchmarks, we address the concern regarding variance by analyzing the Standard Errors provided by the benchmark. As shown in the table below, the observed Standard Errors are low, indicating that the model’s performance is stable and that the reported averages are reliable.
>
> | Model        | ai2d | ChartQA | DocVQA | InfoVQA | RealQA | TextVQA |
> |--------------|------|---------|--------|---------|--------|---------|
> | llama-Siglip | 0.88 | 0.99    | 0.55   | 0.85    | 1.80   | 0.64    |
> | llama-DAVE   | 0.88 | 0.97    | 0.47   | 0.87    | 1.77   | 0.62    |
> | qwen-Siglip  | 0.79 | 0.85    | 0.36   | 0.74    | 1.70   | 0.58    |
> | qwen-DAVE    | 0.78 | 0.76    | 0.35   | 0.76    | 1.80   | 0.59    |
> ***
> **Q2:** there might be potential domain bias, for example overfitting to synthetic chart styles in PlotQA or financial domain layouts in FinTabNet. Might have been useful to detect and discuss such biases.
>
> **A2:** Response: We agree that potential domain bias is an important consideration given the specific nature of datasets like PlotQA or FinTabNet. However, we employ two key strategies to mitigate this risk. First, our architectural design explicitly incorporates a generalist vision encoder (e.g., SigLIP) via ensembling. This ensures that the model retains robust, general-purpose visual representations and does not overfit to the narrow distributions of synthetic document datasets. Second, regarding data composition, datasets such as PlotQA represent only a small fraction of our total pretraining corpus (approx. 2M samples), which is dominated by diverse web screenshots and arXiv PDFs. This diversity naturally dilutes the influence of any single specialized dataset. We will include a more detailed discussion on data composition and bias in the final version.
> ***
> **Q3:** We need to have confidence intervals for the results esp when comparing SigLIP2 to DAVE using Qwen-2.5-7B-Instruct model in Table 1.
>
> **A3:** Please refer to Q1
> ***
> **Q4:** A little more discussion of regression on RealWorldQA and TextVQA would be useful to understand if regressions on general VQA capabilities are real
>
> **A4:** For general VQA tasks, we observe that DAVE outperforms SigLIP 2 on RealWorldQA and TextVQA when paired with Llama 3.2-3B-Instruct. In contrast, SigLIP 2 slightly surpasses DAVE when using Qwen-2.5-7B-Instruct. Interestingly, this trend reverses on MMMU: DAVE performs better with Qwen-2.5-7B-Instruct, while SigLIP 2 leads with Llama 3.2-3B-Instruct. These mixed outcomes suggest that the differences are more likely due to variations in training dynamics and model–task interactions, rather than strong evidence of a regression or advantage in general VQA capability for DAVE. We will include this discussion in the final version.
> ***
> **Q5:** we processed 500K PDF samples from arXiv with an OCR model (Cui et al., 2025), formulating them into additional recognition and grounding tasks. What kinds of post processing was done on the data to ensure high quality examples?
>
> **A5:** Thank you for the question. We first filter arXiv papers to include only those published in venues or journals, ensuring a high-quality source corpus. During post-processing, we remove all bounding boxes with confidence scores below 0.5 to ensure accurate OCR localization. We then randomly resize, crop, and add noise to prevent the model from overfitting to a specific PDF size. Next, we normalize all bounding box coordinates to the range [0, 999] with respect to the new size.
> To control task difficulty, we discard bounding boxes that contain either too few words (e.g., single tokens) or excessively long paragraphs. Finally, from each remaining (category, bounding box, text) tuple, we construct diverse training examples for bbox-to-text, text-to-bbox, and bbox-to-category tasks.
> ***

---

> > ### Comment · Reviewer_gMFC · 2025-11-26
> >
> > Thanks for the clarifications. My questions and concerns are addressed by the authors comments.
> > One followup question on post processing: What was the threshold used for identifying excessively long paragraphs?

---

> ### Author Response · Authors · 2025-11-26
>
> Dear reviewer gMFC,
>
> We are glad that our responses have addressed all your questions, and we truly appreciate your positive evaluation of our work! Regarding the post-processing procedure, we apply a word threshold of 100, separated by space. We qualitatively found that this threshold can capture most captions and short paragraphs in an arXiv paper. To prioritize grounding and save computation, we skip the long paragraphs to avoid performing training on a large portion of the PDF.
>
> Best regards,
>
> Authors

---

> > ### Comment · Reviewer_gMFC · 2025-11-26
> >
> > Thanks for the additional detail. This address all my questions.

---

> > > ### Author Response · Authors · 2025-11-27
> > >
> > > Thank you very much!

---

### Author Response · Authors · 2025-11-26
**General response**

We thank the reviewers for their insightful and constructive feedback! In this work, we introduce DAVE, a vision encoder designed for document understanding and web agentic tasks. We are encouraged that the reviewers recognized our work’s **strong 'breadth of evaluation and ablation'** (`gMFC`) and **'broad applicability'** (`zSTT`). We also appreciate that they highlighted the importance of the document image problem (`Q3WH`) and noted DAVE's ability to **'enhance compatibility across VLM architectures'** (`rHND`) and **'address broad limitations'** (`zSTT`). We are particularly encouraged by the positive scores of **8**(`gMF`) and **6** (`rHND`, **_`rolled back by the ICLR official`_**) following our revisions. We look forward to addressing the remaining concerns to further strengthen the manuscript:

- Contribution and novelty (`Q3WH`, `zSTT`): We emphasize that the novelty of our approach lies in the design of our training framework, not the individual components we apply. This approach differs from vision encoders that are pretrained jointly with self-supervised and supervised objectives. Concretely, the staged combination of self-supervised learning, autoregressive supervised training, and model merging allows us to efficiently leverage large-scale unlabeled data alongside limited high quality annotated data. This is particularly crucial for document and web screenshots, where unlabeled data is abundant but  large scale structures and grounding annotations are bottlenecked by OCR models.

- Choice of generalist encoder (`zSTT`): We conducted an ablation to use a different generalist encoder, and empirical results show that it achieves similar gains to document and web benchmarks. This highlights the robustness of our framework to the choice of generalist encoder.

- Cross-Domain and multilingual performance (`Q3WH`): As suggested, we benchmark DAVE on multilingual documents and general VQAs. Our results show that DAVE remains superior in document VQAs while maintaining performance on general VQAs.

- Regression on general VQA (`gMFC`): We provide an explanation on the potential minor regression of DAVE on general VQA. We highlight the mixed outputs in the experiments and suggest it may be due to training variance instead of strong evidence for regression.

- Analysis on the reason for performance gain (`zSTT`, `rHND`): We provide detailed analysis on the success of DAVE. We attribute the success to the spatial and structural learning by our staged training framework, as evident in Table 3 in the main paper. In addition, We also apply model merging so that DAVE becomes decoder-agnostic, enabling it to be more robust across different VLM architectures. A more comprehensive ablation study of design choices can be found in Figure 2.

**We address below each reviewer’s concern separately and look forward to a constructive discussion.** We will include all updates in the final version of our paper.

---

### Meta-Review · Area_Chair_L9ge · 2026-01-07

**Summary:**

The main concern of this paper is the technical difference to existing methods. It's a valid consideration from reviewers as the paper is a combination of existing techniques (e.g., MAE, Model Merging). AC thought that the combination is non-trivial but also a little bit unnatural. However, the final results seem to be strong, and the paper presented analysis to it.

For other concerns, they have been resolved during rebuttal.

Given this, AC is slightly lean towards positive.

**Reviewer Concerns:**

Remaining Concerns:

1. **Lack of novelty compared with existing methods** (Q3WH, zSTT). Q3WH mainly said that the paper's method is a combination of existing methods. zSTT mainly argue that the paper does not provide enough information to illustrate the effectiveness of such combination. Authors argued with clarification and some experiment support. AC thought that this concern is not fully resolved from reviewer's mind.

2. **DAVE’s performance on RICO-SCA is lower than SigLIP 2** (rHNd). The author acknowledged that ideally the performance should not be lower as DAVE's encoder contains SigLIP features. The author showed that SigLIP do improve the performance of DAVE on semantical understanding. Also, the author also provides some potential reasons of such ineffectiveness but no further experimental evidence is provided.


Resolved Concerns

2. **Limited cross-domain evaluation** (Q3WH) Mostly resolved by additional results in rebuttal.

3. **Missing comparisons with other visual encoder** (zSTT) More results provided in rebuttal.

4. **Whether the method relies solely on the SigLIP2 encoder** (zSTT) Author provides additional results with aimv2.

5. **The variance of performance** (gMFC) Author provides Standard Errors

Todo

6. **Missing details of data post-processing** (gMFC). The author answered the question and reviewer is satisfied. Please make sure to update the final version with such information.

**Reviewer Scores:**

Updated Score: **8 6 6- 6-**

Original Score: **8 4 4 4**

Details:
1. rHNd: original 4, and author claimed that the score has been updated to 6. Unfortunately, there is no discussion to confirm this update.
2. Q3WH: original 4; might be updated to 6 as half of the questions are answered with evidence. Marked as 6-.
3. zSTT: original 4; factual questions are answered. It might remains concern of novelty. Marked as 6-

---

### Decision · Program_Chairs · 2026-01-26

Accept (Poster)